# Human–Animal Bond Generated in a Brief Animal-Assisted Therapy Intervention in Adolescents with Mental Health Disorders

**DOI:** 10.3390/ani13030358

**Published:** 2023-01-20

**Authors:** Maylos Rodrigo-Claverol, Marta Manuel-Canals, Luis Lucio Lobato-Rincón, Natalia Rodriguez-Criado, Maite Roman-Casenave, Elisabet Musull-Dulcet, Esther Rodrigo-Claverol, Josep Pifarré, Yvette Miró-Bernaus

**Affiliations:** 1Primary Health Care Center Bordeta-Magraners, Catalan Institute of Health, 25001 Lleida, Spain; 2Ilerkan Association, 25005 Lleida, Spain; 3Santa Maria University Hospital, 25198 Lleida, Spain; 4Animal-Assisted Intervention Office, Rey Juan Carlos University, Móstoles, 28933 Madrid, Spain; 5San Carlos University Clinical Hospital, 28040 Madrid, Spain; 6Mataró Hospital of Consorci Sanitari del Maresme UHA, 08304 Mataró, Spain; 7ITCAN Dog Assisted Interventions, 08348 Cabril, Spain; 8Primary Health Care Center Primer de Maig, Catalan Institute of Health, 25003 Lleida, Spain; 9Institute for Biomedical Research in Lleida Dr. Pifarré Foundation (IRBLleida), 25198 Lleida, Spain; 10San Juan de Dios Provincia de España, 25001 Lleida, Spain

**Keywords:** animal-assisted therapy, mental health, youth, hospital, adolescent psychiatry, non-pharmaceutical interventions

## Abstract

**Simple Summary:**

Animal-assisted therapy (AAT) is based on the human–animal relationship and the bond that is generated. The effectiveness of AAT will strongly depend on the human–animal bond (HAB) between the patient and the animal. AAT can improve the physical, psychological, and social state of people. Consequently, AAT is particularly relevant in the field of mental health. One hundred and fourteen adolescents with mental health disorders participated in two weekly 1 h group sessions of AAT with therapy dogs. The aim of this study was to assess the degree of HAB generated after short AAT sessions in this young population. The results obtained make it possible to affirm that a short period of time is enough to establish a good HAB between patients and therapy dogs, demonstrating that AAT can be useful in psychiatric hospital acute units.

**Abstract:**

Background: The human–animal bond is crucial for the proper development of animal-assisted therapy (AAT) sessions. However, in specific cases such as in AAT focused on psychiatric patients who are admitted to acute units, there is only a short period of time available to establish this bond. Objectives: To assess the degree of HAB generated after two AAT sessions in adolescents admitted to acute psychiatry units. Methods: A prospective multicenter cohort study aimed at adolescents aged 13 to 17 years admitted to the acute child and adolescent psychiatry unit of three hospitals. Intervention: Two weekly 1 h group sessions of AAT with therapy dogs. Main outcome: The Center for the Study of Animal Wellness Pet Bonding Scale Survey (CSAWPB). Results: One hundred and fourteen adolescents participated. A positive perception of the dog was generated, achieving a good level of bonding and attachment. Conclusions: The results obtained make it possible to affirm that a short period of time is enough to establish a good HAB between patients and therapy dogs. This study aims to contribute to the study of non-pharmacological interventions as a complement to pharmacological treatments in adolescents with mental health disorders.

## 1. Introduction

Animal-assisted therapy (AAT) is a subtype of animal-assisted interventions and is defined, according to the International Association of Human–Animal Interaction Organizations (IAHAIO) [1] as a planned, structured, and documented therapeutic intervention, goal-oriented, and conducted by health or social service professionals as part of their profession. AATs are becoming more widely recognized and scientifically supported as complementary interventions [2] and are innovative resources for the management of situations of functional dependency, psychiatric disorders, behavioral disorders, and care for people with disabilities and special educational needs [3]. For this reason, there has been evidence of an increase in testing and activities with animals that contribute significantly to people’s psychological well-being [4].

AAT is based on the human–animal relationship and the bond that is generated. In the past decade there has been a great interest in research on human–animal interactions (HAI) [5,6]. Several studies about HAI on topics ranging from AAI in special populations, the range of experiences that children have with pets, and the growing university infrastructure supporting HAI research have been published [7]. Likewise, the connection between human–animal health, and the unique and enduring bond between dogs and humans, is attracting significant attention in different disciplines, with a focus on the concept of the human–animal bond (HAB) [8].

It has been argued that the effectiveness of AAT will strongly depend on the generated HAB between the patient and the animal, together with their relationships and bonds with the AAI technician or the canine guide and any other health professionals involved during the therapy [9]. These interactions might provide a context that improves communication and self-confidence, reduces disease symptoms, and improves quality of life, with the dog acting as a social catalyst and social support [10,11]. Likewise, some authors indicate that participants’ expectations and the presence of animals may affect their perception of the therapist and their attitude towards the disclosure of personal information, which are elements that affect the probability of success [12]. With this HAB factor in mind, AAT aims to develop bonding skills in patients on the basis of HAI [13]. Attachment styles and behaviors indicative of maintaining contact during AAT are taken as additional indicators of HAB strength [14].

Among the beneficial effects of AAT, these supportive animal therapies are effective in improving social skills, communication skills, and coping abilities [15,16] reducing psychological distress and depression [15], reducing the perception of loneliness [17], decreasing anxiety, improving mood, aiding independent living, and enhancing empathic skills [16], decreasing blood pressure, and increasing the neurochemicals associated with relaxation and bonding [17]. Consequently, AAT is particularly relevant in the field of mental health. However, there is still a disconnection in the field of mental health and HAB, as they are not mentioned in most research and clinical training curricula [18]. In addition, among mental health disorders patients, those admitted to acute units spend a short period of time in the hospital, thus limiting the time for interventions. This raises the question whether this period is enough for establishing an adequate HAB and therefore to provide an effective AAT.

In recent years, rates of mental disorders have increased in childhood and adolescence, including common mental disorders such as schizophrenia and mania, addictions, suicidal behavior, and personality disorders [19]. In addition, it is evident that the presence of child and adolescent mental health problems can lead to increased economic, social, or healthcare costs. Mental disorders in young people have a high percentage of persistence and chronicity into adulthood [19].

Despite the availability of various approved psychosocial treatments for children and adolescents, a current overview of treatments reveals that there is still much to be done [20,21,22]. In this sense, the incorporation of AAT can be a motivating and enabling element of therapy in adolescents.

In view of the above, the objectives of this study were to assess acute psychiatric children and adolescents’ perception of the dog’s presence and the degree of attachment with the dog after a short period of time (14 days) with only two sessions.

## 2. Materials and Methods

### 2.1. Design and Participants

A prospective multicenter cohort study was performed, aimed at adolescents aged 13 to 17 years admitted to the acute child and adolescent psychiatry unit of three hospitals participating in the study: the Santa Maria University Hospital of Lleida, the Consorci Sanitari del Maresme Hospital of Mataró, and the Niño Jesús University Children’s Hospital of Madrid.

The inclusion criteria were: attendance at two AAT sessions, willingness to participate in the study on a voluntary basis, submission of the information sheet, and signature of the informed consent form (participant and legal guardian). Participants were excluded if they stated in the initial interview that they had an allergy to or fear of dogs, a history of aggression towards animals, re-admissions who had already participated in the study, or if, once informed, the patient and/or legal guardian did not want to participate in the study.

### 2.2. Selection Process

Participation was offered to all of the patients admitted to the units participating in the study and who met all the inclusion criteria and none of the exclusion criteria. The adolescents and their legal guardians were previously informed, were given the information sheet, and, if they accepted, were asked to sign the informed consent forms.

### 2.3. Variables

#### 2.3.1. Response Variables

Center for the Study of Animal Wellness Pet Bonding Scale (CSAWPBS) [23]:

The Center for the Study of Animal Wellness Pet Bonding Scale (CSAWPBS) version validated in Spanish was filled out by the participants themselves after the intervention. The Spanish version was created using the back translation validation system.

CSAWPBS is a scale that aims to measure the participants’ perception of the dog’s presence, feelings of reciprocity between the participant and the dog, and the participants’ degree of attachment to the dog. It presents statements such as “the pet dog likes me” and “the pet dog knows when I feel happy”.

It is a 28-item scale where the participant rates the statement on a 5-point Likert scale from 1 = always true, to 5 = never true. Possible scores range from 28 to 140, with lower scores indicating an increased perception of bonding: 28 = always true, 56 = usually true, 84 = neutral (neither true nor false), 112 = rarely true, and 140 = never true.

This instrument has three subscales to measure bonding factors: unconditional acceptance by the canine companion, feeling of reciprocity for the visit, and attachment to the animal [24]. The subscales: unconditional acceptance is composed of four items (1, 6, 21, 24); reciprocity is composed of ten items (2, 4, 5, 11, 14, 15, 16, 25, 26, 27); and companion attachment is composed of 14 items (3, 7, 8, 9, 10, 12, 13, 17, 18, 19, 20, 22, 23, 28). The rating for statement 17 was marked in reverse because it is worded as a negative statement and is different from the other 27. The results of the subscales were scaled from 1 = always true to 5 = never true.

#### 2.3.2. Independent Variables

Age at inclusion (in years); gender (boy or girl); hospital (Santa Maria University Hospital of Lleida, the Consorci Sanitari del Maresme Hospital of Mataró, and the Niño Jesús University Children’s Hospital of Madrid); diagnosis of depression (yes or no); diagnosis of psychosis (yes or no); diagnosis of behavioral disorder (yes or no); diagnosis of eating disorder (yes or no); and pet ownership (yes or no).

### 2.4. Intervention

A total of two one-hour group sessions were held at the hospitals’ own facilities, on a weekly basis and for two consecutive weeks. The groups had a maximum of 10 participants.

All of the sessions were planned, drafted, and agreed upon in advance by the teams’ professionals. In each session, specific objectives were worked on by means of different group dynamics. Six dynamics were presented that could be chosen at the discretion of each team of professionals. The reason for presenting six different dynamics that shared the same objectives was due to the possibility of there being participants who were still hospitalized after more than 2 weeks; in this way, they could continue to attend the sessions (even if they were already out of the study) without having to repeat the same sessions. The dynamics are specified in the Appendix A.

### 2.5. Human Resources

Each hospital’s occupational therapist and/or psychologist participated in the sessions, along with the AAI technician from each of the entities that carried out the intervention in each hospital. There was a referring psychiatrist for the project at each center.

### 2.6. Animal Resources

The intervention included six therapy dogs, selected for having a suitable character and appropriate aptitudes, with a calm, docile, and obedient temperament, and a training that enriched the sessions. Specifically, two golden retrievers (one male and one female), one male labradoodle, and three mixed-breed dogs weighing 19–25 kg (one male and two females) participated, averaging 4 years of age.

The animals completed a specific training process as therapy dogs, always with positive training techniques and habituation to the hospital environment in order to be able to easily adapt to different situations. The dogs were always accompanied by the AAI technicians, who are trained in ethology and dog training and always had animal welfare in mind. Each technician lives with his/her therapy dogs in order to promote the bond between the professional guide for the animal and the dog itself. In addition, the standards of the Animal-Assisted Intervention Office of Rey Juan Carlos University of Madrid were met, both in terms of animal welfare and zoonosis prevention. All of the animals were periodically checked by a veterinarian to ensure that they were in good health and that basic standards of vaccination, anti-parasite treatment, and hygiene were met. Each hospital’s access protocol and hygiene measures for hand washing, change of clothes, and surface disinfection/cleaning were followed. Liability insurance was also arranged.

### 2.7. Analysis

The sample has been described according to sociodemographic variables and the bond between the participants and the dog (according to the answers to the CSAWPBS test). Averages and standard deviations were used for numerical variables and absolute and relative frequencies were used for categorical variables. The different results were compared according to the sex of the participants and according to primary diagnoses (depression, psychosis, behavioral and eating disorders). To evaluate the differences between the groups, Student’s *t*-tests were used for numerical variables and chi-square tests were used for categorical variables. In cases in which the expected count in some cells was lower than five, a Fisher’s exact test was used instead of a chi-square test. Statistical significance has been established for *p* < 0.05.

### 2.8. Ethical Considerations

The study protocol was initially approved by the Research Ethics Committee (REC) of the Hospital Universitario Arnau de Vilanova of Lleida, REC identification code 2080. Followed by the REC of the Hospital Infantil Universitario Niño Jesús of Madrid, code R-0042/19, and the Consorci Sanitari of Maresme Hospital of Mataró, Code 51/19. Confidentiality was maintained in accordance with the European Union’s General Data Protection Regulation 2016/679. The study followed the principles of the Declaration of Helsinki [25]. Animal welfare and zoonosis prevention protocols were applied. Liability insurance was arranged.

## 3. Results

### 3.1. Participant Characteristics

The participants in this study formed a total sample of 114 people, they had a mean age of 14.9 (1.61), and 84.2% were girls. Of the sample, 45.6% came from Lleida, 27.2% came from Mataró, and the remaining 27.2% came from Madrid.

The most frequent diagnoses were depression and eating disorders and there were no significant differences by gender in terms of age or diagnoses (Table 1). There were also no significant differences in age based on diagnosis, except for participants with psychosis who had a mean age of 16.6 (0.55) years (*p* < 0.001) (data not shown).

### 3.2. Results of Participant–Dog Bond

Regarding the bond generated between the participants and the dogs during the intervention, the CSAWPBS scale total was 56.4 (18.1) (the results are specified in the Appendix A). The results of the total sample in the subscales were: unconditional acceptance 1.61 (0.55), reciprocity 2.12 (0.73), and attachment 2.06 (0.76) (Table 2). No significant differences were found in the total scale nor in the subscales based on gender (Table 2), whether they had a pet (Table 3), or diagnosis (Table 4).

Regarding pet ownership, it should be noted that participants who live with a pet (i) would like the visiting dog to come to their home (*p* = 0.044), (ii) think that the animal accepts them as they are (*p* = 0.012), and (iii) think that the dog helps them to forget their problems (*p* = 0.038) (the results are specified in the Appendix A). When the results were analyzed by sex, the items “the visiting dog tries to make me feel comfortable and calm” (*p* = 0.027) and “the dog’s visit makes me feel happy” (*p* = 0.024) were more positively rated by boys than by girls.

If we analyze the results according to diagnosis, we can observe that participants diagnosed with psychosis emphasize that the dog’s visits are not boring (*p* < 0.001) and that the dog helps them to feel safe (*p* = 0.008). Participants with behavioral disorders favorably perceive that the visiting dog is always happy to see the participants (*p* = 0.017), they trust the dog (*p* = 0.027), and they think that they make the dog feel happy (*p* = 0.017) as well as making the dog feel better (*p* = 0.014). Likewise, participants with eating disorders have the sensation that they make the dog feel happy (*p* = 0.021), perceive that the visiting dog knows when they feel bad (*p* = 0.017), and the dog makes them feel safer (*p* = 0.008). The results are specified in the Appendix A.

## 4. Discussion

The objective of this study was to determine the HAB generated after a brief intervention using AAT in adolescents with mental health disorders.

A total of 114 adolescents were included in the study, of whom 84.2% were girls; this high percentage of females corresponds with the proportion of hospital admissions, which may be attributed to the fact that the most frequent diagnoses were depression and eating disorders, which are more prevalent in girls [26].

The bond generated between the participants and the dog during the intervention was high, with a total CSAWPBS scale score of 56.4 (18.1), which means “usually true”. When the bond is assessed by subscales, unconditional acceptance is the factor that had the most influence on the bond with a value of 1.61 (0.55), which translates to “always true” and represents maximum acceptance. We experience unconditional acceptance when we are accepted and loved just as we are. Under these conditions, we do not feel the need to adapt to an external expectation of how we should be. In this sense, the dog does not criticize, accepting the participants without discriminating against anyone, and does not judge [10,27] fundamental aspects that help us to feel accepted.

Regarding the reciprocity subscale, the global score was 2.12 (0.73), which can be judged as “usually true”. Reciprocity is one of the manifestations of pro-social behavior, it is a process of exchange where mutual benefit is sought. It implies that, if someone makes a concession to us, we are obliged to respond with another concession of our own. If we do not, we feel guilty. The socialization process plays an important role in the development of this need for reciprocity [28].

The results obtained in the reciprocity subscale are also judged as “usually true” (2.06 (0.76)). Attachment is an intense and lasting affective bond that develops and consolidates between two individuals, through their reciprocal interactions, and that provide security, comfort, and protection [9,29]. Thus, the feeling of security and affection towards the animal perceived by the adolescents constitutes a very important factor for being able to generate the feeling of attachment [30]. The authors agree that animals have a set of facilitating factors for attachment, as therapy animals behave spontaneously; they are always available for interaction, they are non-judgmental, they provide unconditional love, and are loyal and affectionate. These elements corroborate that a climate of security and trust can be fostered and help to establish and further consolidate the therapeutic relationship [30,31].

Taking into account the level of the bond generated between the participant and the dog and the valued results in the main agents of the bond, the unconditional acceptance, reciprocity, and attachment generated between patient and animal, they may favor a context in which a higher level of therapeutic alliance can be established with the patient regardless of their diagnosis. Therapeutic alliance can be considered a good predictor of therapy success [32]. A similar study by Mezza et al. [33], which assess the changes observed during the AAT sessions in relation to the human–animal bond, increasing therapeutic alliance, depth of elaboration, and smoothness of the treatment sessions, highlights the essential role of such dimensions taking place in ATT. These findings also seem to suggest a positive outcome of the treatments, and the pivotal role of such variables is documented as the strongest predictor of outcomes in psychotherapy [33].

Although 21.9% of the participants usually lived with a pet, this study has shown that the fact of living with a pet does not add any significant value to the participants’ perceptions and does not seem to influence the results of the AAT, in line with previous studies [34]. For this reason, some studies indicate that further research is needed for the investigation of the effects of companion animals on human health and wellbeing, because the kind of bonding that the person has with the companion animal might acquire different meanings depending on its psychological significance [35].

The presence of the HAI in AAT is critical to the participants’ having reported positive experiences in terms of enjoying the interactions, bonding, gaining emotional support, and facilitating social interaction [36,37]. However, there is a lack of evidence of similar interventions that assess the level of HAB, making our study with a short AAT intervention one of the few to report findings regarding this variable. Many studies evaluate the efficacy of interventions only in terms of attachment and the therapeutic relationship between participants and professional or guardians/parents of participants and professional [38,39].

One question frequently raised by AAI professionals is the optimal duration of the intervention in order to obtain results. Thus, comparing a similar study by Avila et al. [3], which consisted of a single short session of AAT, and our study with two sessions of AAT, they turned out to be sufficient to obtain results. However, in a systematic review that analyzed several studies, there was a great variety of results regarding the duration of the interventions but it was lacking in results regarding the longevity of the beneficial effects of the interventions [40]. Thus, our study could provide interesting data by discovering that with only two sessions, a bond with the animal has already been generated, as well as a positive perception in both the participants and the professionals.

In view of the above, it is thought that a brief AAT intervention in adolescents with mental health disorders admitted to mental health hospitals can be an effective complementary tool for improving the biopsychosocial well-being of young people. However, more studies are needed to be able to verify the behavior, attitudes, and bonding generated during AAT in other settings, encompassing the needs of the youth population.

### Limitations

First of all, it is important to highlight that there is not a well-defined cutoff point related to a positive HAB, so further research regarding this topic should be performed. However, our results show that the HAB established during the AAT sessions is not neutral and there is a high agreement with the different statements of the CSAWPBS test, indicating a globally positive HAB.

The absence of information on potentially important variables such as the degree of severity of medical diagnoses or comorbidities can also be considered a weakness. Although, no differences have been found in the results when compared according to diagnoses.

The scale and questionnaires handed out involve the subjectivity of the responding participants, and this may vary depending on their mood on that particular day. However, it is difficult to assess attachment with any other methodology.

Finally, it is not known whether the positive effects were sustained beyond the period studied, to what extent, or for how long after the activity. It would therefore be interesting to consider evaluating the long-term effect in future studies, by handing out the questionnaires again after several weeks or months. This should be discussed in the broadest context possible. Future research directions may also be highlighted.

## 5. Conclusions

The results obtained make it possible to affirm that after two AAT sessions, a positive perception of the dog was generated, achieving a good level of bonding and attachment.

A dog may be a motivating and enabling element in therapy for adolescent patients. This study aims to contribute innovation and research to an under-studied area that can be useful in an adjunctive manner for pharmacological treatments in patients with mental health disorders.

## Figures and Tables

**Table 1 animals-13-00358-t001:** Description of the sample.

	All Participants (N = 114)	Boys (N = 18)	Girls (N = 96)	Statistic (*t*-Value or Chi-Squared Value)	*p*-Value (Boys vs. Girls)
Sociodemographic variables					
Sex (Girls)	96 (84.2%)	-	-	-	-
Age	14.9 (1.61)	14.1 (2.14)	15.0 (1.46)	−1.69	0.106
Region				-	0.434
Lleida	52 (45.6%)	6 (33.3%)	46 (47.9%)		
Mataró	31 (27.2%)	7 (38.9%)	24 (25.0%)		
Madrid	31 (27.2%)	5 (27.8%)	26 (27.1%)		
Owns a pet	25 (21.9%)	4 (22.2%)	21 (21.9%)	-	1
Clinical variables					
Depression	49 (43.0%)	8 (44.4%)	41 (42.7%)	0.00	1
Psychosis	5 (4.39%)	1 (5.56%)	4 (4.17%)	-	0.584
Behavioral disorder	13 (11.4%)	4 (22.2%)	9 (9.38%)	-	0.124
Eating disorder	40 (35.1%)	4 (22.2%)	36 (37.5%)	-	0.328
Borderline personality disorder	2 (1.75%)	0 (0.00%)	2 (2.08%)	-	1
Bipolar disorder	2 (1.75%)	0 (0.00%)	2 (2.08%)	-	1
Autism spectrum disorder	4 (3.51%)	1 (5.56%)	3 (3.12%)	-	0.502
Emotional disorder	2 (1.75%)	0 (0.00%)	2 (2.08%)	-	1
Obsessive–compulsive disorder	2 (1.75%)	1 (5.56%)	1 (1.04%)	-	0.292

**Table 2 animals-13-00358-t002:** Participant–dog bond with the total sample and according to gender.

CSAWPBS	All Participants (N = 114)	Boys (N = 18)	Girls (N = 96)	*t*-Value (Boys vs. Girls)	*p*-Value (Boys vs. Girls)
Bond with the dog (total score)	56.4 (18.1)	51.4 (17.7)	57.3 (18.1)	−1.302	0.205
Unconditional acceptance subscale	1.61 (0.55)	1.62 (0.53)	1.61 (0.56)	0.114	0.91
Reciprocity subscale	2.12 (0.73)	1.93 (0.70)	2.15 (0.73)	−1.235	0.228
Attachment subscale	2.06 (0.76)	1.82 (0.70)	2.10 (0.76)	−1.503	0.145

Total score interpretation: 28 = always true, 56 = usually true, 84 = neutral (neither true nor false), 112 = rarely true, and 140 = never true. Subscales interpretation: 1 = always true, 2 = usually true, 3 = neutral (neither true nor false), 4 = rarely true, and 5 = never true.

**Table 3 animals-13-00358-t003:** Participant–dog bond according to pet ownership.

CSAWPBS	No (N = 89)	Yes (N = 25)	*t*-Value (Yes vs. No)	*p*-Value
Human–dog bond	57.4 (19.2)	52.7 (13.0)	−1.439	0.156
Unconditional acceptance subscale	1.63 (0.58)	1.53(0.45)	−0.966	0.399
Reciprocity subscale	2.15 (0.77)	2.00 (0.59)	−1.049	0.299
Attachment subscale	2.10 (0.80)	1.89 (0.559	−1.488	0.142

Total score interpretation: 28 = always true, 56 = usually true, 84 = neutral (neither true nor false), 112 = rarely true, and 140 = never true. Subscales interpretation: 1 = always true, 2 = usually true, 3 = neutral (neither true nor false), 4 = rarely true, and 5 = never true.

**Table 4 animals-13-00358-t004:** Participant–dog bond according to diagnosis.

	Depression				Psychosis				Behavioral Disorder				Eating Disorder			
CSAWPBS	No (N = 65)	Yes (N = 49)	*t*-Value (Yes vs. No)	*p*-Value	No (N = 109)	Yes (N = 5)	*t*-Value (Yes vs. No)	*p*-Value	No (N = 101)	Yes (N = 13)	*t*-Value (Yes vs. No)	*p*-Value	No (N = 74)	Yes (N = 40)	*t*-Value (Yes vs. No)	*p*-Value
Human–dog bond	55.4 (17.3)	57.7 (19.2)	0.669	0.505	56.6 (18.2)	50.8 (15.7)	−0.810	0.459	57.1 (17.9)	50.8 (19.6)	−1.106	0.287	54.6 (18.5)	59.6 (17.1)	1.457	0.149
Unconditional acceptance subscale	1.55 (0.47)	1.69 (0.65)	1.315	0.192	1.61 (0.56)	1.60 (0.60)	−0.045	0.966	1.62 (0.57)	1.58 (0.46)	−0.282	0.781	1.61 (0.60)	1.61 (0.47)	0.010	0.992
Reciprocity subscale	2.11 (0.73)	2.13 (0.74)	0.170	0.865	2.13 (0.74)	1.88 (0.58)	−0.922	0.402	2.15 (0.73)	1.90 (0.77)	−1.100	0.289	2.02 (0.73)	2.29 (0.71)	1.936	0.056
Attachment subscale	2.01 (0.72)	2.11 (0.81)	0.713	0.477	2.07 (0.76)	1.83 (0.61)	−0.842	0.441	2.08 (0.74)	1.83 (0.85)	−1.024	0.323	2.00 (0.77)	2.16 (0.73)	1.121	0.266

Total score interpretation: 28 = always true, 56 = usually true, 84 = neutral (neither true nor false), 112 = rarely true, and 140 = never true. Subscales interpretation: 1 = always true, 2 = usually true, 3 = neutral (neither true nor false), 4 = rarely true, and 5 = never true.

## Data Availability

The data presented in this study are available on request from the corresponding author.

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
