# Peer review of "Human–Animal Bond Generated in a Brief Animal-Assisted Therapy Intervention in Adolescents with Mental Health Disorders"

_animals, 2023, doi:10.3390/ani13030358_

Round 1

Reviewer 1 Report

The manuscript is of great interest to the Journal and it was a pleasure reading it.

It is well-structured and well-written.

Below are only a couple of things that might improve the paper:

Line 91: “mental health patients” is not very clear as an expression.

Line 139 should better explain the independent variables.

Finally, the paper might be improved by taking into account the articles of Mezza et al. on Process Evaluation of Animal Assisted Therapies with Children, the paper of Mezzalira et al. on Affiliative bonds and cruel behavior in childhood interspecific relationships, and the paper by Scandurra et al. on Animal-Assisted Education Intervention with Dogs to Promote Emotion Comprehension in Primary School Children.

Author Response

Dear reviewer,
Please find attached the revised version of our manuscript entitled: “Human-Animal Bond Generated in a Brief Animal Assisted Therapy Intervention in Adolescents with Mental Health Disorders”. 

We would like to thank the editor and the reviewers for their nice comments about our manuscript. We hope that this new version would be suitable for publication in Animals Journal.

We have enclosed  the cover letter for response your comments and the new manuscript. 

Yours sincerely,

Dra. M. Rodrigo-Claverol

Response to Reviewer 1 Comments

The manuscript is of great interest to the Journal and it was a pleasure reading it.

It is well-structured and well-written.

We thank the reviewer for his/her nice comments.

Point 1: Line 91: “mental health patients” is not very clear as an expression.

Response 1:  We agree with the reviewer too. The expression is not clearand it is necessary to clarify the concept that defines patients as "mental health disorders patients". (line 94)

Point 2: Line 139 should better explain the independent variables.

Response 2:  We apologize if the reviewer did not find if the independent variables have not been described in detail. 

“Age at inclusion (in years); gender (boy or girl); hospital (Santa Maria University Hospital of Lleida, the Consorci Sanitari del Maresme Hospital of Mataró, and the Niño Jesús University Children's Hospital of Madrid); diagnosis of depression (yes or no); diagnosis of psychosis (yes or no); diagnosis of behavioral disorder (yes or no); diagnosis of eating disorder (yes or no), pet ownership (yes or no).” (line 148-152)

Regarding this clarification, we have also modified the variables exposed in the tables, for example; girl instead of woman, boy instead of man, and pet ownership instead of "pet variable". These changes are also displayed in tables 1, 2 and 3.

Point 3: Finally, the paper might be improved by taking into account the articles of Mezza et al. on Process Evaluation of Animal Assisted Therapies with Children, the paper of Mezzalira et al. on Affiliative bonds and cruel behavior in childhood interspecific relationships, and the paper by Scandurra et al. on Animal-Assisted Education Intervention with Dogs to Promote Emotion Comprehension in Primary School Children.

Response 3: We thank the reviewer for these appreciations. The articles that the reviewer has proposed are of great interest and help in the discussion of our results.

“A similar study by Mezza et al [33], which assess the changes observed during the AAT sessions in relation to the human-animal bond, increasing therapeutic alliance, depth of elaboration, and smoothness of the treatment sessions, highlight the essential role of such dimensions taken in place in ATT. These findings also seem to suggest a positive outcome of the treatments, and it is documented the pivotal role of such variables as the strongest predictors of outcomes in psychotherapy [33]. “ (line 297-302)

“For this reason, some studies indicate that further research is needed as to the investiga-tion of the effects of companion animals on human health and wellbeing. Because the kind of bonding that the person has with the companion animal might acquire different meanings depending on its psychological significance[35].”  (line 306-309)

Reviewer 2 Report

this manuscript reports results of a 2-session interaction with a therapy animal on the perceived human-animal bond among adolescents in a psychiatric hospital. It's an interesting piece and it adds to the growing literature on AAT. Before I can recommend it for publication, I have some suggestions, that fall somewhere between minor and major revisions.

L23 – simple summary. First word is TAA? What is that? Do you mean AAT? Even if so, write it out for the first time it’s used.

L27 – ‘mental diseases’ is probably not the right term. Suggest changing to ‘mental disorder’, which is the term used by the WHO https://www.who.int/news-room/fact-sheets/detail/mental-disorders Same with the title. Mental disorder might be better than mental health problems

L31 – end of simple summary – suggest adding one sentence with practical implications.

Abstract. There are ; throughout where . would be more appropriate (e.g., after ‘this bond; Objectives…’ on L35.

L36 ‘Prospective multicentre cohort study’ is not a sentence, and neither is the one immediately after.

L42-44 – this sentence could be phrased more concisely. It is very awkward as is. Also, it mentions that this is an understudied area, but AAT generally is not. Be specific about what is not studied much.

Intro

L59 – TAA is here again. Check this carefully throughout, please.

L59-61 – what is ‘unprecedented’ about the research happening in this area? That is unclear. Also, there is a * at the end of the sentence, but I can’t find what it is referring to.

L67-72 – all of the claims in these lines are made as if they are based on empirical evidence, but the three citations referenced are all lit reviews/position papers. None of these claims has been empirically proven yet, although they all make sense. I suggest changing the phrasing to say that ‘it has been argued that…’ rather than making these claims as a statement of fact. The evidence base is not at the point where we can definitely say that social support and social catalyst effects are responsible for any observed benefits of AAT. Same with the impact of the HAB on AAT. It is likely that it affects it, but we don’t know for sure. The rest of this para is written better, in a more nuanced way.

L86-89 are unclear, and the citation may be in the wrong place, but I’m not certain.

Generally, it would be helpful to have a bit more justification of the adolescent population. Why would teens, in particular, benefit from AAT? Even though adolescents are the population of interest, they are barely given any attention in the Intro.

Methods

Generally good.

Analysis – have you considered comparing the first vs second visits on the HAB measure? It’s possible that the HAB is established very early on, as early as the first visit, or maybe it develops over time. It would be useful information.

L131-137 – the total scale score interpretations are provided, but not the subscale scores. What is the range for the subscales?

L136 – change to past tense; instead of ‘will be marked’, change to ‘was marked’

L175 – evaluation of the effectiveness of the treatment – where is this described in the results? And it is not mentioned in the materials section, either. How was this determined?

Results

L197 – since these participants were underage, I don’t think ‘women’ is the appropriate term here. Suggest changing to ‘girls’.

Tables – as above, ‘men’ and ‘women’ would not be the correct terms for people under 18. Suggest changing to ‘boys’ and ‘girls’. For all tables, I suggest adding the t-value or chi-square value in addition to the p-values, according to standard scientific convention. These are important when reporting statistical results.

Table 3 – suggest changing caption to ‘Participant-dog bond according to pet ownership’, rather than ‘…’pet variable’’.

Tables 3 and 4 – I don’t see the rationale for including every individual item in the analysis if you’ve already done the subscales. Why not just report the subscales, as in Table 2? The whole point of having subscales is to reduce the number of variables required for analysis. Table 4 in particular is overwhelming with the amount of data being reported. Just reporting the subscales will make it more manageable for the reader to understand. Finally, I suggest adding a brief footnote to these tables to remind the reader what a score on the total scale and subscales represents. On L127-137 in the methods, the info is there for the total scale, but not the subscales, which would be useful. But briefly adding it to the tables will mean that the reader can easily interpret the results without having to go back to the methods.

L217-230 – suggest removing these if you also remove the results from the table, as suggested above. If you really want to keep them, I suggest changing L217 ‘Regarding the “pet” variable’ to say merely ‘Regarding pet ownership…’

Discussion

L234-236 – This makes sense, but does it also roughly correspond with the hospital admissions more generally? Is it mostly girls who are in the hospitals anyway, or is it an even split? If it’s an even split or if it’s mostly boys, then this needs to be mentioned as a limitation because it will reduce the generalisability of the results.

References

Seem fine, but the list includes the number in the reference itself (e.g., reference 2 says ‘2.  2.NAME, TITLE..’, so there’s an extra 2.

Author Response

Dear reviewer,
Please find attached the revised version of our manuscript entitled: “Human-Animal Bond Generated in a Brief Animal Assisted Therapy Intervention in Adolescents with Mental Health Disorders”. 

We would like to thank the editor and the reviewers for their nice comments about our manuscript. We hope that this new version would be suitable for publication in Animals Journal.

We have enclosed  the cover letter for response your comments and the new manuscript. 

Yours sincerely,

Dra. M. Rodrigo-Claverol

Response to Reviewer 2 Comments

We thank the reviewer for his/her comments and contributions. We are sure that they are of great help to improve the article.

Point 1: L23 – simple summary. First word is TAA? What is that? Do you mean AAT? Even if so, write it out for the first time it’s used.

Response 1:  We thank the reviewer for this appreciation. According to the reviewer’s comment we have clarified the meaning of the abbreviation AAT as “Animal assisted therapy”.

“Simple Summary: Animal assisted therapy (AAT) is based on the human-animal relationship and the bond that is generated. The effectiveness of AAT will strongly depend on the generated HAB between the patient and the animal.” (lines 23-25)

Point 2: L27 – ‘mental diseases’ is probably not the right term. Suggest changing to ‘mental disorder’, which is the term used by the WHO https://www.who.int/news-room/fact-sheets/detail/mental-disorders Same with the title. Mental disorder might be better than mental health problems

Response 2:  We agree with the reviewer comment, and we also agree that “mental disorders” are more adequated.

“Consequently, AAT is particularly relevant in the field of mental health. One hundred fourteen adolescents with mental disorders participated in two weekly 1-hour group sessions of AAT with therapy dogs.” (lines 26-27)

Point 3: L31 – end of simple summary – suggest adding one sentence with practical implications.

Response 3:  We thank the reviewer for the suggestion. We have added practical implications that have been used in the study.

“The results obtained make it possible to affirm that a short period of time is enough to establish a good HAB between patients and therapy dogs, demonstrating that AAT can be useful in psychiatric hospital acute units.” (lines 30-32)

Point 4: Abstract. There are ; throughout where . would be more appropriate (e.g., after ‘this bond; Objectives…’ on L35.

Response 4:  We apologize for the mistake, we have already rectified the punctuation marks in the different points of abstract.

“.Background: Human-Animal Bond is crucial for the proper development…” (line 33)

“.Objectives: To assess the degree of HAB…” (line 36)

“.Methods: A prospective multicenter cohort study…” (line 37)

“.Conclusions: The results obtained…” (line 42)

Point 5: L36 ‘Prospective multicentre cohort study’ is not a sentence, and neither is the one immediately after.

Response 5: We apologize for the mistake, we have already rectified the sentence.

Methods: A prospective multicenter cohort study aimed at adolescents aged 13 to 17 years admitted to the Acute Child and Adolescent Psychiatry Unit of three hospitals” (line 37)

Point 6: L42-44 – this sentence could be phrased more concisely. It is very awkward as is. Also, it mentions that this is an understudied area, but AAT generally is not. Be specific about what is not studied much.

Response 6: We thank the reviewer for this indication. We have not expressed it clearly, so we have modified the sentence in order to make it better understood.

“This study aims to contribute to the study of non-pharmacological interventions as a complement to pharmacological treatments in adolescents with mental health disorders.” (lines 44-47)

Point 7: Intro: L59 – TAA is here again. Check this carefully throughout, please.

Response 7:  We have rectified all TAA abrebiatios of all manuscript, and we have change for AAT (TAA is in Spanish version).

Point 8: L59-61 – what is ‘unprecedented’ about the research happening in this area? That is unclear. Also, there is a * at the end of the sentence, but I can’t find what it is referring to.

Response 8:

On the one hand, we also agree that “unprecedented” expression is not clear. So, we have changed for “great interest”.

On the other hand, we have rectified * (is not a reference).

“AAT is based on the human-animal relationship and the bond that is generated. In the past decade there has been a great interest  in research on human-animal interactions (HAI) [5,6] (lines 59-61)

Point 9: L67-72 – all of the claims in these lines are made as if they are based on empirical evidence, but the three citations referenced are all lit reviews/position papers. None of these claims has been empirically proven yet, although they all make sense. I suggest changing the phrasing to say that ‘it has been argued that…’ rather than making these claims as a statement of fact. The evidence base is not at the point where we can definitely say that social support and social catalyst effects are responsible for any observed benefits of AAT. Same with the impact of the HAB on AAT. It is likely that it affects it, but we don’t know for sure. The rest of this para is written better, in a more nuanced way.

Response 9:

We appreciate the reviewer’s suggestions; we have modified it in the text.

“It has been argued that the effectiveness of AAT will strongly depend on the generated HAB between the patient and the animal, together with their relationships and bonds with the AAI technician or canine guide and any other health professionals involved during the therapy [9]. These interactions might provide a context that improves communication and self-confidence, reduces disease symptoms, and improves quality of life, with the dog acting as a social catalyst and social support [10,11].” (lines 67-72)

Point 10: L86-89 are unclear, and the citation may be in the wrong place, but I’m not certain.

Response 10:

We apologise for this mistake. This reference is wrong, so we have eliminated.

Point 11: Generally, it would be helpful to have a bit more justification of the adolescent population. Why would teens, in particular, benefit from AAT? Even though adolescents are the population of interest, they are barely given any attention in the Intro.

Response 11:

We apologize for not clarifying this aspect in the previous version of the manuscript. We have added new references that justified this field.

“In recent years, rates of mental disorders have increased in childhood and adolescence, including common mental disorders such as schizophrenia and mania, addictions, suicidal behavior, and personality disorders [19]. In addition, it is evident that the presence of child and adolescent mental health problems can lead to increased economic, social or healthcare costs. Mental disorders in young people have a high percentage of persistence and chronicity into adulthood [19].

Despite the availability of various approved psychosocial treatments for children and adolescents, a current overview of treatments reveals that there is still much to be done [20-22]. In this sense, the incorporation of AAT can be a motivating and enabling element of therapy in adolescents.” (lines 91-100)

Point 12: Analysis – have you considered comparing the first vs second visits on the HAB measure? It’s possible that the HAB is established very early on, as early as the first visit, or maybe it develops over time. It would be useful information.

Response 12:

It is an aspect that could have been very interesting to evaluate. Unfortunately the test was not passed at these time points and cannot be compared.

Point 13: L131-137 – the total scale score interpretations are provided, but not the subscale scores. What is the range for the subscales?

Response 13:

We thank the reviewer for this appreciation. According to the reviewer’s comment we have specified the subscale scores.

“This instrument has three subscales to measure bonding factors: unconditional acceptance by the canine companion, feeling of reciprocity for the visit, and attachment to the animal [24]. The subscales: unconditional acceptance is composed of four items (1,6,21,24); reciprocity is composed of ten items (2,4,5,11,14,15,16,25,26,27); and companion attachment is composed of 14 items (3,7,8,9,10,12,13,17,18,19,20,22,23,28). The rating for statement 17 was marked in reverse because it is worded as a negative statement and is different from the other 27. The results of the subscales were scaled from 1 = always true, to 5 = never true.” (lines 136-143)

Point 14: L136 – change to past tense; instead of ‘will be marked’, change to ‘was marked’

Response 14:

Thank for the aprecciation, we have correct de mistake.

“The rating for statement 17 was marked in reverse because it is worded as a negative statement and is different from the other 27. The results of the subscales were scaled from 1 = always true, to 5 = never true (lines 140-143)

Point 15: L175 – evaluation of the effectiveness of the treatment – where is this described in the results? And it is not mentioned in the materials section, either. How was this determined?

Response 15:

The evaluation of the effecteviness of the treatment is not specified, it has been a mistake. We have clariefied this section.

“The sample has been described according to sociodemographic variables and the bond between the participants and the dog (according to the answers to the CSAWPBS test). “(line 190-192)

Point 16: L197 – since these participants were underage, I don’t think ‘women’ is the appropriate term here. Suggest changing to ‘girls’.

Response 16:

We agree with the reviewer and we have changed for “girls” term.

“The participants in this study formed a total sample of 114 people, they had a mean age of 14.9 (1.61) and 84.2% were girls.” (lines 211-213)

Point 17: Tables – as above, ‘men’ and ‘women’ would not be the correct terms for people under 18. Suggest changing to ‘boys’ and ‘girls’. For all tables, I suggest adding the t-value or chi-square value in addition to the p-values, according to standard scientific convention. These are important when reporting statistical results.

Response 17:

On the one hand,  regarding terms which are used for people under 18, we have  modified the variables exposed in the tables and all manuscript, for example; girl instead of woman, boy instead of man, and pet ownership instead of "pet variable". These changes are also displayed in tables 1, 2 and 3.

On the other hand, we thank the suggestion regarding the t and chi-squared values. We have added them in the tables. We have detected that for most of the results in Table 1, a Fisher’s exact test was used instead of a Chi-square test, in this case we have not added the chi-squared value, since the statistic for the Fisher’s test is the p-value itself. We have specified the criteria used for performing a Fisher’s test in the Analysis subsection.

Point 18: Table 3 – suggest changing caption to ‘Participant-dog bond according to pet ownership’, rather than ‘…’pet variable’’.

Response 18:

We thank the reviewer for the suggestion. We have rectified the caption as reviwer has proposed.

Point 19: Tables 3 and 4 – I don’t see the rationale for including every individual item in the analysis if you’ve already done the subscales. Why not just report the subscales, as in Table 2? The whole point of having subscales is to reduce the number of variables required for analysis. Table 4 in particular is overwhelming with the amount of data being reported. Just reporting the subscales will make it more manageable for the reader to understand. Finally, I suggest adding a brief footnote to these tables to remind the reader what a score on the total scale and subscales represents. On L127-137 in the methods, the info is there for the total scale, but not the subscales, which would be useful. But briefly adding it to the tables will mean that the reader can easily interpret the results without having to go back to the methods.

Response 19:

We think it is a great sugestion. We also agree that reporting the subscales make it more manageable for the reader to understand. So, we have eliminated individual items in tables 3 and 4, which now it can be found in suplementary material.

Moreover, we have added brief footnote to these tables to remind the reader what a score on the total scale and subscales represents

Point 20: L217-230 – suggest removing these if you also remove the results from the table, as suggested above. If you really want to keep them, I suggest changing L217 ‘Regarding the “pet” variable’ to say merely ‘Regarding pet ownership…’

Response 20:

We thank the reviewer for this suggestion. According to the reviewer’s comment we have changed all mentions about pet ownership.

“Regarding the pet ownership, it should be noted that participants who live with a pet” (lines 243)

Point 21: Discussion: L234-236 – This makes sense, but does it also roughly correspond with the hospital admissions more generally? Is it mostly girls who are in the hospitals anyway, or is it an even split? If it’s an even split or if it’s mostly boys, then this needs to be mentioned as a limitation because it will reduce the generalisability of the results.

Response 21:

We apologize if the reviewer did not find clear enough this aspect. We have clarified this aspect, so in the discussion it is shown the percentage of famales that corresponr with the propotion of hospital admisions.

“A total of 114 adolescents were included in the study, of whom 84.2% were girls; this high percentage of females corresponds with the proportion of hospital admissions, which may be attributed to the fact that the most frequent diagnoses were depression and eating disorders, which are more prevalent in girls [26]” (lines 262-265)

Point 22: References: Seem fine, but the list includes the number in the reference itself (e.g., reference 2 says ‘2.  2.NAME, TITLE..’, so there’s an extra 2.

Response 22:

We apologize for the mistake; we have modified all references with extra number.

Round 2

Reviewer 2 Report

The authors have done a great job with this second version of the ms and I'm happy to accept it for publication.